# Skin Microbial Community Associated to Strawberry Disease in Farmed Rainbow Trout (*Oncorhynchus mykiss* Walbaum, 1792)

**DOI:** 10.3390/microorganisms12010217

**Published:** 2024-01-21

**Authors:** Alda Pardo, Alejandro Villasante, Jaime Romero

**Affiliations:** 1Laboratorio de Biotecnología de Alimentos, Instituto de Nutrición y Tecnología de los Alimentos (INTA), Universidad de Chile, El Líbano 5524, Santiago 7830489, Chile; alda.pardo@ug.uchile.cl (A.P.); alejandro.villasante@inta.uchile.cl (A.V.); 2Cooperative Program for Aquaculture (Ph.D.), Universidad de Chile, Universidad Católica del Norte, Pontificia Universidad Católica de Valparaíso, Valparaíso 2340025, Chile; 3Facultad de Medicina Veterinaria y Agronomía, Universidad de Las Américas, Santiago 7500000, Chile

**Keywords:** strawberry disease, skin microbiota, *Candidatus* Midichloria, rainbow trout, aquaculture, red mark syndrome

## Abstract

Aquaculture plays a crucial role in addressing the growing global demand for food. However, diseases associated with intensive aquaculture practices, especially those affecting the skin, can present significant challenges to both fish health and the industry as a whole. Strawberry disease (SD), also known as red-mark syndrome, is a persistent and non-lethal skin condition observed in Rainbow Trout (*Oncorhynchus mykiss*) in the United States and various European countries. SD is a nonlethal skin condition of an unclear etiology that affects rainbow trout reared in freshwater close to the harvest period. We used a RNA-based approach to examine active microbiota in the SD skin lesions and compared to non-injured skin. Our results, based on using 16S rRNA gene next-generation sequencing, showed that the skin microbiota was dominated by the phyla Firmicutes, Proteobacteria, and Actinobacteria. The comparisons of the skin microbiota between injured and non-injured samples showed differences in the alpha diversity (Fisher index) and beta diversity metrics (ANOSIM). At the genus level, both *Pseudomonas* and *Candidatus* Midichloria were highlighted as the most abundant taxa detected in samples obtained from fish affected with strawberry diseases. In contrast, the most abundant taxa in non-injured skin were *Escherichia-Shigella*, *Streptococcus,* and *Pseudoalteromonas*. In conclusion, our study on SD revealed distinct differences in the microbiota composition between skin lesions and non-injured skin. This is the first description of microbiota associated with SD-injured skin samples using an RNA approach.

## 1. Introduction

In recent years, skin diseases, including warm water strawberry disease (WWSD), formerly known as Strawberry disease (SD), have emerged as major concern in *Oncorhynchus mykiss* farming in Europe as well as North and South America [1,2,3,4]. Strawberry disease is a skin disorder affecting rainbow trout in freshwater. Although the disease does not result in significant mortalities, it does cause considerable economic losses to the trout industry due to the downgrading of the fish at the time of harvest.

In Chile, in pan-sized trout farms located in the Los Lagos and Araucanía regions, the appearance of inflammations and skin lesions has been observed, mainly on the flanks of the fish and on the dorsal and ventral surfaces [4]. This condition is observed in trout close to harvest (300–600 g). Histopathological, microbiological, and virological analyses were carried out by Chilean researchers, and they have determined that these lesions are characteristic of SD, first described in the United States [4,5,6]

The farming units experiencing an SD case do not exhibit associated mortality or changes in behavior. However, they do manifest evident lesions, resulting in a morbidity rate of 80% [3,4]. Due to these characteristics in the compromised batches, farms cannot be harvested, as current national regulations prohibit the reception of fish with lesions in processing plants. The Chilean pan-sized trout industry is concentrated in small-scale aquaculture enterprises where the economic losses related to SD are very significant. A similar situation has been recently reported in Perú, where farms from two locations (Lima and Junin) presented outbreaks of similar signs [7]. However, those authors described the pathology as Red Mark Syndrome (RMS). Previous reports have suggested that Red Mark Syndrome (RMS), also known as cold-water strawberry disease (CSWD), and Strawberry Disease (SD) are the same condition observed in both different geographic locations and water temperatures [8,9].

Strawberry disease (SD) was first reported in America in the state of Washington and characterized as focal, non-suppurative dermatitis with extensive ulceration and the infiltration of mononuclear cells [10]. The etiology of SD has yet to be confirmed, despite efforts to isolate and cultivate the responsible pathogen. Various techniques, including microbiological, histological, immunohistological, and transmission electron microscopy analyses, have been employed to detect the responsible pathogen; however, conclusive results have not been achieved. However, it is believed to have a bacterial origin, as indicated by the research [11,12]. Ferguson et al. [11] conducted a study involving a limited number of fish affected by Strawberry Disease (SD) and RMS in the US and the UK, respectively. They proposed a potential link between RMS in Scottish fish and *Flavobacterium psychrophilum*, identified through a polymerase chain reaction (PCR). Subsequent research by Lloyd et al. [13] revealed a strong association between SD lesions and the presence of a Rickettsia-like organism (RLO) 16S rRNA sequence. When examined using quantitative PCR (qPCR) and immunohistochemistry (IHC), RMS samples from the UK yielded results similar to those obtained for SD samples from the US, based on the RLO 16S rRNA sequence [8].

The integumentary system in fish, comprising the skin and mucus, serves as both physical and biochemical barriers that delineate the organism from its environment [14,15]. These surfaces harbor a diverse and intricate bacterial community, potentially contributing to fish homeostasis [16,17]. The composition of the skin-associated microbiota in teleosts is reportedly influenced by external factors, including the physicochemical parameters of the aquatic environment (such as temperature, pH, and salinity). In the case of farmed fish, additional factors like diet, treatments, handling, and stress also play a role [18,19]. The skin microbiota has been associated with various physiological functions in fish, including the immune response, gas exchange, osmoregulation, and excretion, contributing to the dynamic equilibrium between the host and the bacterial community [20,21].

An imbalance in the skin mucus bacterial community is often linked to reduced microbial diversity and a higher prevalence of pathogenic or opportunistic bacteria [22,23]. Therefore, a comprehensive understanding of the composition of the skin microbiota is essential for unraveling its role in fish health and diseases, particularly those related to the skin. In this context, the aim of this study is to elucidate the composition of the skin-associated microbiota in rainbow trout by contrasting the microbiota linked to Strawberry Disease (SD) lesions with that of healthy trout. To accomplish this objective, an RNA-based approach was employed, leveraging its capability to enhance the recovery rate of template molecules by approximately 1000-fold, specifically in terms of 16S rRNA molecules per bacterial cell [24]. This methodology enables the identification and characterization of the active bacterial population within the sample.

## 2. Materials and Methods

### 2.1. Samples

Rainbow trout skin samples were obtained from specimens reared in a commercial trout farm located at Región de Los Lagos, Chile, specializing in the grow-out phase of pan-sized rainbow trout. This facility displayed a condition of Strawberry Disease (SD) in five pre-harvest raceways. The selected raceway to be sampled had no previous antibiotic treatment in the last 6 months and no handling procedures in the last 30 days. Skin samples were obtained using specimens of rainbow trout reared in one raceway showing 50% morbidity. It corresponded to a batch of eggs of national origin. The average cultivation temperature for this batch had been 9.5 °C. Trout were captured using clean dip nets (kept in a 200 ppm iodine solution) assigned exclusively for this purpose, alternating their use for each fish to ensure they remained in the disinfectant solution for 10 min before being reused. The average weight of the sampled fish was 383 g, 6 corresponded to healthy trout and 6 corresponded to specimens showing skin lesions. 

### 2.2. Skin Sampling

The sampling procedure occurred in a necropsy room following aseptic conditions. To euthanize the fish, each trout was placed in a bucket with an overdose (200 mg/L) of tricaine methanesulfonate (MS 222, Merck, Darmstadt, Germany), adhering to the recommendations outlined in the “Guide for the Care and Use of Laboratory Animals of the National Institutes of Health” and the animal ethics guide of INTA, University of Chile [24]. The sampling was described by [24] using scalpel for vertical incision from dorsal to pectoral fin, obtaining 0.5 mm × 0.8 mm epithelial tissue. The skin samples were carefully placed in 1.5 mL Eppendorf tubes containing 700 microliters of TRIzol. Additionally, separate Falcon tubes filled with 10% buffered formalin were utilized for histological purposes. The samples designated for molecular analysis were subjected to a cooling phase, as described [24]. Meanwhile, the samples intended for histological examination were consistently kept at room temperature.

### 2.3. Histological Examination

The skin samples, fixed in buffered formalin, underwent dehydration in an ascending alcohol series and were subsequently immersed in xylene for subsequent embedding in paraffin. Subsequently, 5 μm sections were obtained, rehydrated through a descending alcohol series, and ultimately stained with hematoxylin and eosin [25]. A Leica DM2000 light microscope (Leica Microsystems, Wetzlar, Germany) was employed for their examination and histological characterization.

### 2.4. RNA Extraction and PCR Amplification

For RNA extraction, 250 mg samples of skin were weighed and thawed at room temperature. The samples were processed as described [24]. Once the RNA elutions were obtained, cDNA was synthetized using random primers and a Reverse Transcription System Promega kit (Promega, Madison, WI, USA). PCR amplifications were performed as previously described [25]. PCR products were purified using a QIAquick^®^ PCR Purification Kit (Qiagen, Hilden, Germany). Libraries were sequenced on the paired end Illumina platform Hiseq PE250 adapted for 300-bp paired-end reads at the CD Genomics (http://www.cd-genomics.com; accessesd on 15 December 2023) as described [24].

### 2.5. Bioinformatic Analysis

Reading quality assessed with FastQC 0.11.8. Duplicate sequences and chimeras removed using DADA 2 in RStudio 4.1. Taxonomic assignment done using Silva database, creating OTU and taxonomy tables with specified packages [25].

## 3. Results

### 3.1. Histological Analysis

The lesions caused by SD, as depicted in Figure 1, were examined using a histological approach to describe and compare the structural differences with healthy skin. In SD lesions, the structural changes in the dermis result in the loss of stratum integrity and thickening due to lymphocyte infiltration, extending from the dermis to the muscular layer. This is in stark contrast to the observation of typical structures in histological skin sections without lesions, where no significant alterations are present. These descriptions are illustrated in Figure 1, highlighting the contrast between fish without lesions and those with lesions. The skin samples with SD injuries can be classified as severe according to [26], as they include lesions larger than 2 cm, exhibiting pronounced redness and hemorrhages, along with a moderate-to-severe exudate. Additionally, there is a noticeable loss of scales and the frequent erosion of the epidermis.

### 3.2. Sequencing Results

The sequencing results, considering all samples sent for NGS (*n* = 12), yielded 977,234 initial reads and 755,687 final reads after various filtrations performed in the DADA 2 pipeline. The average number of reads per sample was 62,716 +/− 7779. Despite variations in the number of reads per sample, a high level of coverage (0.99) was achieved using rarefaction, as demonstrated in Appendix A, which displays the obtained rarefaction curves.

### 3.3. Bacterial Diversity of Skin Microbiota

To determine the alpha diversity, Chao 1 indices based on sample richness and the Fischer index considering both richness and distribution were employed. The results indicate that, at the genus level, there is higher alpha diversity in the composition of the bacterial community in skin samples without lesions. Figure 2A represents the Fischer index as a boxplot. Each boxplot illustrates the diversity distribution of a group within the ‘Condition’ class. The estimated statistical significance yielded a *p*-value of 0.002, with a test statistic [*t*-test] of −4.056. The results obtained using the Chao 1 index showed higher richness at the genus level in the condition without lesions, as depicted in Figure 2B. The boxplot illustrates the diversity distribution of a group within the ‘Condition’ class. The estimated statistical significance resulted in a *p*-value of 0.0051, with a test statistic [*t*-test] of −3.620. These data show a significantly higher level of bacterial diversity in the skin of trout without lesions.

To determine Beta diversity between conditions at the OTU level, the Weighted Unifrac distance matrix was employed and graphically represented using Principal Coordinates Analysis (PCoA) (Figure 3). This methodology takes into account both the richness and abundance of taxa present in each sample to establish if there are significant differences between conditions. Permutational Multivariate Analysis of Variance (PERMANOVA) was utilized to assess significant differences between conditions. The results indicate significant differences between the bacterial community associated with SD lesions (injury) and the bacterial community associated with healthy skin (*p*-value of 0.003; FDR value of 0.003).

### 3.4. Microbial Composition of the Skin Microbiota

Skin microbiota was dominated by bacteria assigned to the phyla Proteobacteria, Firmicutes, and Actinobacteriota, which jointly accounted for over 90% of the total bacterial communities across all the samples. The highest relative abundance in skin samples without SD lesions (non injury) is composed of the Proteobacteria with 67.38%, followed by Firmicutes with 19.66%, and Actinobacteriota with 8.88%. In samples with SD lesions (injury), the highest relative abundance is also given by the Phylum Proteobacteria, but with a higher percentage of 88.77%, followed by Firmicutes and Actinobacteriota, but with lower percentages of 4.81% and 2.7%, respectively, as shown in Figure 4.

At the genus level, skin samples without SD lesions (non-injury) exhibited the highest relative abundance in the genera *Escherichia_Shigella* at 20.76%, followed by *Pseudoalteromonas* with 15.09%, and third in abundance was the genus *Streptococcus* at 9.61%. In contrast, in the condition with SD lesions (injury), the highest relative abundance observed was in the genus *Pseudomonas* with 36.18%, followed by the genus *Candidatus* Midichloria with 19.66%, and in third place, the genus *Acinetobacter* with 9.34%, as shown in Figure 5. It is noteworthy that the abundance of the genus *Candidatus* Midichloria and *Pseudomonas* in skin samples without SD lesions (non-injury) was 0.46% and 1.92%, respectively.

The sequence of the ASV assigned to *Candidatus* Midichloria (265 bp) was compared to the sequence from the database (Genbank) to check assignation and also their identities from different sources were compared (Appendix A). The table shows that the ASV sequence was highly similar, showing a 99–100% matched identity (0 to 2 mismatches) to sequences retrieved from trout and trout parasites and named as a Midichloria-like organism (MLO). However, this identity similarity dropped to 95% (>11 mismatches) when compared to other sequences retrieved from ticks associated to terrestrial animals.

In order to analyze the effect size of differences in abundance and composition at the genus level, the Lefse tool was employed. This tool highlights the taxonomic representation of statistically and biologically consistent differences between two conditions. The results obtained, in terms of the differences in the taxonomic distribution of groups of trout with skin and without SD lesions, show that the three genera with the most significant differences in abundance in the lesion condition, compared to the non-lesion condition, were *Pseudomonas*, *Candidatus* Midichloria, and *Acinetobacter*. Conversely, the genera *Pseudoalteromonas*, *Streptococcus*, and *Staphylococcus* exhibited the most significant differences in abundance in the non-lesion condition compared to the lesion condition, as depicted in Figure 6. 

## 4. Discussion

The Strawberry disease, identified in North and South America, and Red Mark Syndrome, currently present in Europe and Asia, are described as the same epithelial disorder [8,11], causing significant economic losses in pan-sized trout farming. This condition not only jeopardizes the harvest of affected fish due to food safety regulations, which prohibit the processing of fish with wounds, but also requires a period for either natural reversal or antibiotic treatment, ranging from 8 to 4 weeks, respectively. To avoid any confounding effects associated with culture conditions, including inadequate food, stress, increased housing density, etc., we sampled both strawberry disease-affected fish and unaffected fish from the same raceway. This enabled us to compare between the skin-associated microbiota of trout with and without lesions under the same environmental and handling conditions as well as similar feeding and dietary management.

With regard to the histological characterization of wounds, our findings agree with previous works (Figure 1 and Appendix A). An inflammatory response was observed, which, depending on the degree of progression of the lesion, was extended from the dermis to the hypodermis, accompanied by a significant loss of scales and leukocyte infiltration [4,5,26,27]. The observed inflammatory response indicates the activation of an innate immune response in the lesions [27,28,29]. This type of reaction constitutes a defensive response, as the inflammatory process aims to protect the host by eliminating the pathogens [14,17]. Additionally, the inflammatory response helps to remodel the histological environment to facilitate leukocyte migration, activates chemical mediators, and deploys antimicrobial mechanisms in the affected tissue [14,30]. 

Overall, in our study, the bacterial composition associated with the lesions differs from that associated with lesion-free skin, as evidenced by the PERMANOVA (Figure 5) and the Lefse analyses (Figure 6). Further, we observed different taxa associated either with the condition with lesions or the condition without lesions. This suggests that the stability of the protective barrier composed of mucus and skin-associated microbiota was compromised, leading to an imbalance (dysbiosis) that allowed the entry of pathogens into the epithelial tissue [30,31,32,33]. 

In terms of the microbial composition, our findings are consistent with prior research either utilizing DNA or RNA extraction, where *Proteobacteria* emerged as the dominant phylum in the epithelial mucus of teleosts, irrespective of the aquatic environment (i.e., marine or freshwater). Notably, Legrand et al. (2020) [34], based on an RNA-based approach, emphasized the prevalence of the *Proteobacteria* phylum in the active microbiota of the external mucosal surface in *Seriola lalandi*. Similarly, Lokesh and Kiron (2016) [35], using DNA extraction, reported *Proteobacteria* as the phylum of the highest relative abundance in the skin-associated microbiota during the transition from freshwater to seawater in Atlantic salmon (*Salmo salar*). In open-flow freshwater-cultured rainbow trout, Lowrey et al. [36] observed *Proteobacteria* as the phylum with the highest relative abundance, exceeding 50%. 

Regarding composition at the genus level, the high relative abundance of *Pseudomonas* and *Candidatus* Midichloria were greater in the SD-injured skin samples. The *Pseudomonas* genus belongs to *Proteobacteria* and are notable for their ubiquity, being found in both terrestrial and aquatic environments, including freshwater and marine habitats [37,38]. This genus has a high capacity to utilize a broad range of nutrients, showing great adaptability to the environment compared to other bacteria genera [37], and thus making them opportunistic pathogens in a wide range of animal species. Remarkably, this genus has been established as part of the skin-associated microbiota of fish, producing toxins against other bacteria, such as *Flavobateria,* and thus constituting a potential antagonist to flavobacteriosis [31,39]. Indeed, this might be used as an inhibitory competition strategy by *Pseudomonas* against skin pathogens in the context of a disruption of skin integrity in salmonids, such as trout. Hence, these antecedents support our findings where *Pseudomonas* bacteria can take advantage of the organic matter and epithelial destruction found in skin wounds, providing a rich source of carbon and nitrogen for their growth and survival. 

Previous studies have reported the presence of *Candidatus* Midichloria in both SD and RMS lesions [8,26,27,40]; however, these works have relied on the PCR technique, implying some limitations to establish in-depth comparisons of the relative abundances of dominant bacterial genera. In contrast, in our study, a more comprehensive and functional characterization of the associated microbiota as well as the potential role in the skin lesions was performed using a high-throughput sequence. Here, we report a significantly greater relative abundance of RLO, *Candidatus* Midichloria, in severe SD-injured skin (Figure 5) compared to the unaffected skin samples. This is in line with a previous study on the possible etiological agent of RMS and SD using molecular techniques, especially in the absence of a culture system for RLO [2,10]. To identify a potential bacterial agent responsible for skin disease (SD), 16S ribosomal DNA libraries were constructed from SD lesion samples and healthy skin samples. Within these libraries, a highly similar 16S ribosomal DNA sequence to members of the order Rickettsiales was identified, and was present in three lesion libraries with a prevalence of 1%, 32%, and 54%, respectively. Further, this sequence was not detected in any library from healthy tissues. Interestingly, in the present study, the sequence of the ASV assigned to *Candidatus* Midichloria (Appendix A) was highly similar to the sequences retrieved from the library prepared by Lloyd et al. (2008) [2]. Subsequently, Lloyd et al. (2021) [13] developed a quantitative Taqman PCR assay (qPCR) to quantify RLO in relation to the severity of the lesions. The qPCR results revealed a higher number of RLO sequence copies in severe lesions compared to fewer copies in mild lesions, suggesting that RLO is the etiological agent of Strawberry Disease (SD). Therefore, taking into consideration the above-mentioned antecedents, it appears that *Candidatus* Midichloria could be the etiological agent of SD; however, to probe this statement, an appropriate pathogen challenged-based experimental design is required. 

The genus *Candidatus* Midichloria belongs to the order *Rickettsiales* and the family *Midichloriaceae*. Some species within this genus are symbionts of the common tick, residing in the ovary cells of female ticks [41]. In terrestrial environments, RLOs have been associated with diseases in vertebrates, and their primary vectors are typically arthropods, such as ticks [42]. In aquatic environments, RLOs have been linked to diseases in invertebrates, including oysters, white shrimp, and red abalone and vertebrates, including horse mackerel, carp, and sea bass, with limited information on associated vectors [43,44]. Specifically, in the case of SD, an RLO present in skin lesions was similar to an RLO found in arthropods and closely phylogenetically related to RLO detected in amoebas, leeches, and trematodes [45,46,47]. Protozoa carrying RLOs, such as *Ichthyophthirius multifiliis*, commonly known as ICH, cause the white spot disease [47]. It is noteworthy that in Chile, ICH was a common pathology in trout fry, especially in lake culture centers and open-flow fish farming, such as the one from which our samples were obtained. Interestingly, the sequence of the ASV assigned to *Candidatus* Midichloria (265 bp) showed a high similarity to the RLO observed in *I. multifiliis* (Appendix A) [46]. Further, *I. multifiliis* (ICH), when exposed to fish affected by RMS, becomes positive for RLO, suggesting that the bacterium can be acquired and carried by the protozoan, at least transiently.

The condition of SD or RMS in rainbow trout appears to be horizontally transmissible, and according to the current literature [4,5,45], a bacterial infection is considered the potential etiology, with Rickettsia-like organisms deemed the most probable candidate. Additional research revealed a strong and positive correlation between fish affected by RMS and an RLO identified as a Midichloria-like organism (MLO), belonging to the Midichloriaceae family and the order Rickettsiales [4,46]. Hence, an MLO is considered the potential cause of RMS or SD [4,5,44,45,46,47]. MLOs were identified in splenic impression smears, presenting morula-like structures characteristic of RLOs. These structures were observed in the cytoplasm of splenic macrophages in rainbow trout affected by RMS [48,49,50]. Transmission electron microscopy (TEM) further validated the presence of intracytoplasmic microorganisms resembling *Rickettsiales* within macrophages, fibroblasts, and erythrocytes [5,26,48,49,50]. Therefore, considering the observed inflammatory response, which has the potential to induce alterations in the skin layers, our findings point towards a disruption in the stability of the protective barrier constituted by mucus and skin-associated microbiota. These result in an unbalanced microbiota (dysbiosis) that diminishes bacterial diversity and fosters the proliferation of *Candidatus* Midichloria, an MLO associated with SD and RMS. Additionally, *Pseudomonas*, an aerobic bacterium renowned for its ability to utilize and break down a diverse array of organic compounds as carbon and energy sources, is implicated. *Pseudomonas* is also known for secreting degrading enzymes, such as proteases and lipases, which have the potential to exacerbate skin damage, not only predisposing to secondary pathogen infections, including bacteria and oomycetes (i.e., *Saprolegnia parasitica*), but also negatively affecting osmoregulation due to ion and water exchanges that can be regulated by creating barriers at the skin surface (e.g., scales and mucus layers) that limit permeability in fish [51]. 

Finally, despite both SD and RMS having been suggested as the same fish skin disorder, they show some epidemiological differences that go beyond the mere geographical locations of the outbreaks since their respective outbreaks have been reported to occur at different water temperatures as well. Indeed, SD has been described to occur when water temperatures exceed 16 °C, whilst Red Mark Syndrome has been reported at water temperatures of less than 15 °C [11,12,52]. While skin disorders can result from biotic factors like fungi, protozoa, bacteria, and viruses, as well as abiotic factors such as allergies or sunburn, SD has not been linked to predisposing factors like management practices, facility conditions, diet, or water conditions. Nonetheless, stress has been proposed as a potential trigger or exacerbating factor for this disorder in cultured trout [52]. Therefore, based on the above-mentioned antecedents, it is possible that this disease might be limited only to trout under culture conditions, with no manifestation in the wild trout population, and thus not implying a pathogenic role in wild trout. 

## 5. Conclusions

RNA-based approaches can be highly valuable for studying the microbiota communities of samples with low bacterial counts, such as injured fish skin. They are particularly useful when aiming to describe the active microbiota. Our investigation into Strawberry Disease (SD) has unveiled notable distinctions in the microbiota composition between skin lesions and non-injured skin. This marks the initial characterization of the microbiota associated with SD-injured skin samples through RNA analysis, highlighting a substantial prevalence of *Candidatus* Midichloria.

## Figures and Tables

**Figure 1 microorganisms-12-00217-f001:**
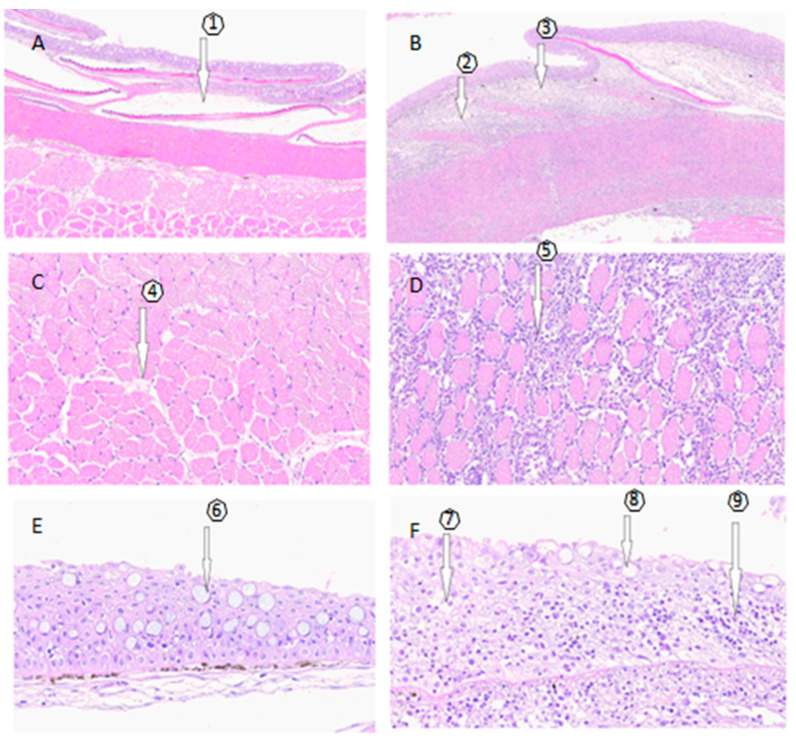
Histological examination. (**A**,**C**,**E**) depict the histology of skin samples without SD lesions, while (**B**,**D**,**F**) show samples of skin with SD lesions. (**A**) No alterations from the epidermis to the muscular layer, with the presence of scales (1). (**B**) Severe diffuse dermatitis in the compact and spongy dermal stratum with a predominance of lymphocyte infiltration. Arrows indicate the partial absence of scales, (2) and (3). (**C**) Without lesions and no findings in the muscular layer (4). (**D**) At the muscular level, severe diffuse myositis is observed (5), along with mild multifocal degeneration and necrosis of the muscle. (**E**) Skin sample without lesions, no findings, goblet cells (6). (**F**) Lesioned skin sample, showing dermatitis in the compact and spongy dermal layer (7), loss of continuity of the epidermis (8), and a predominance of lymphocytes in the compact and spongy dermal layer (9).

**Figure 2 microorganisms-12-00217-f002:**
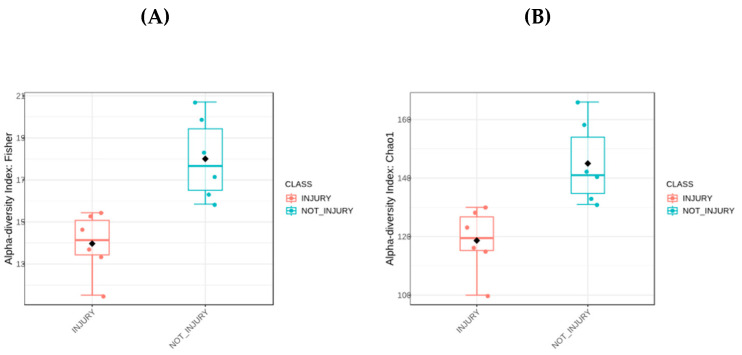
Alpha diversity analyses. (**A**) Boxplot of Fisher index. (**B**) Boxplot of Chao index. Black dot: Mean; horizontal bar in box: Median.

**Figure 3 microorganisms-12-00217-f003:**
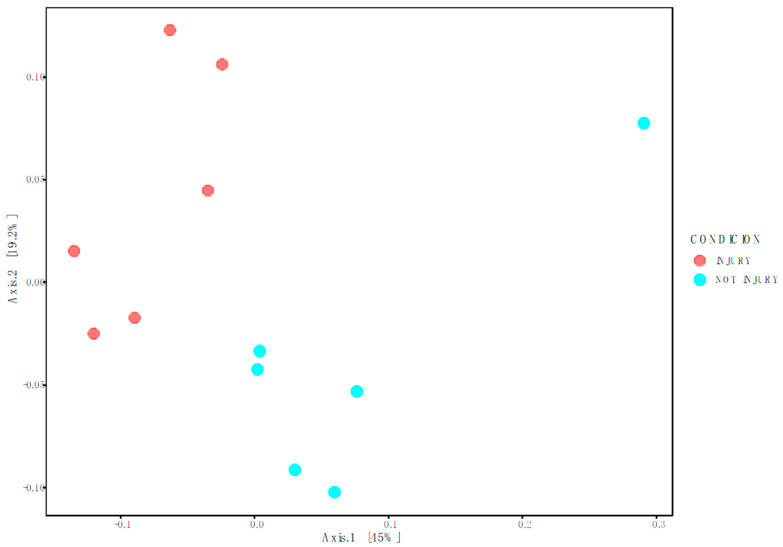
Weighted Unifrac represented in Principal Coordinates Analysis (PcoA).

**Figure 4 microorganisms-12-00217-f004:**
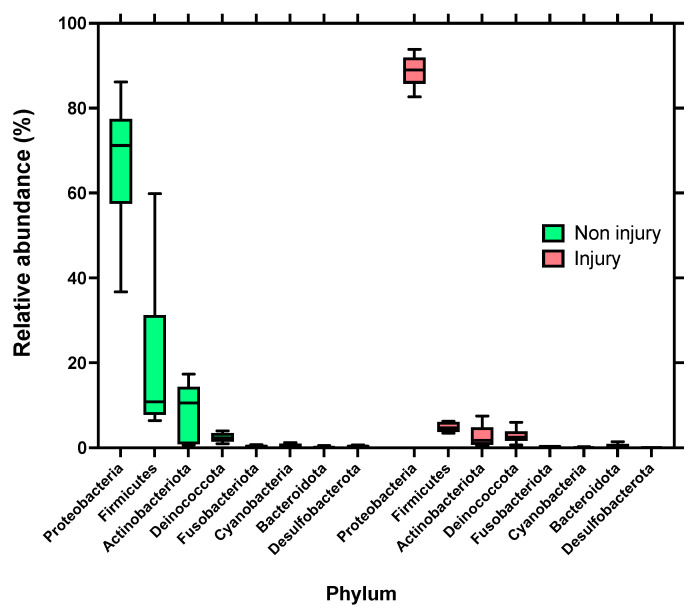
Relative abundance distributions of bacterial phyla assignable to amplicon sequence variants (ASVs) from skin microbiota of rainbow trout (*Oncorhynchus mykiss*). Left side skin samples without SD lesions (non injury); right side: skin samples with SD lesions (injury).

**Figure 5 microorganisms-12-00217-f005:**
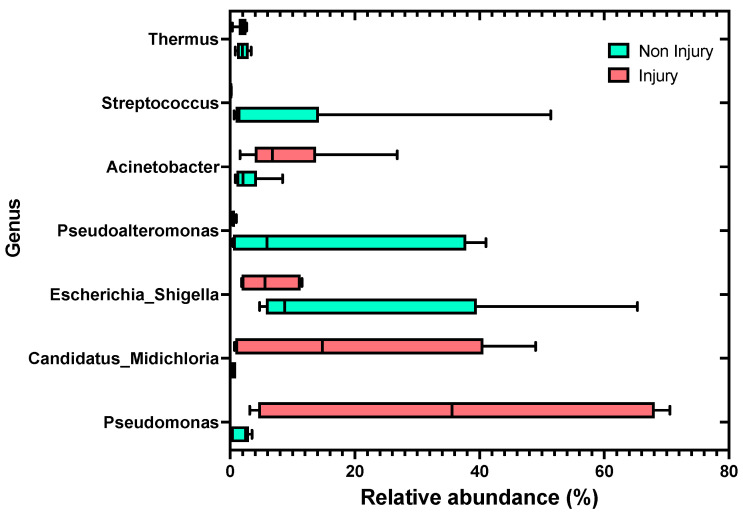
Relative abundance distributions of bacterial genera assignable to amplicon sequence variants (ASVs) from the skin microbiota of rainbow trout (*Oncorhynchus mykiss*), comparing skin samples without SD lesions (non-injury) to skin samples with SD lesions (injury).

**Figure 6 microorganisms-12-00217-f006:**
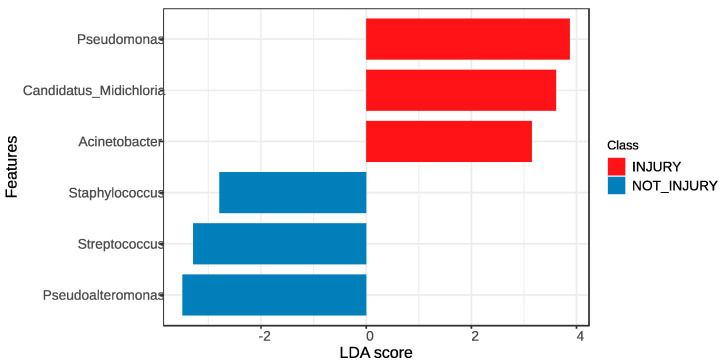
LEfSe was used to determine the statistical significance and the effect size of the differential abundance of taxa between skin samples without SD lesions (non-injury) and skin samples with SD lesions (injury).

## Data Availability

Data are contained within the article and Appendix A.

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
