# Peer review of "Skin Microbial Community Associated to Strawberry Disease in Farmed Rainbow Trout (Oncorhynchus mykiss Walbaum, 1792)"

_microorganisms, 2024, doi:10.3390/microorganisms12010217_

Round 1
Reviewer 1 Report
Comments and Suggestions for Authors
The manuscript is devoted to the study of the microbiota of rainbow trout, sampled in places of skin lesions during the development of the so-called Strawberry Disease. The authors used a variety of and generally adequate research methods, including molecular analysis of sampled species. The results obtained were analyzed using modern methods of statistical analysis and illustrated.
In general, the results obtained in the study are in the trend of modern studies of this type of fish diseases that arise during the industrial method of fish cultivation.
Of greatest interest, in my opinion, is the region where the research is being conducted (South America), where there is clearly not enough such work being carried out.
Overall, the manuscript deserves publication in the journal Microorganisms, although it requires some corrections.
The main comments are as follows:
Text should be checked for misspellings, see for example genus name in line 35. When a species is first mentioned, it is necessary to also indicate the authors and year: Oncorhynchus mykiss (Walbaum, 1792).
In the abstract and text there is clearly an excessive number of abbreviations for the name of the disease mentioned. Only in the abstract the abbreviation SD is given three times. It is enough to do this once in the abstract and once at the first mention in the text.
The abstract is unjustifiably verbose; it contains information, for example, the place of which is in materials and methods (an indication of weight of fish). The text of the abstract should be significantly reduced, leaving only the most essential.
The main drawback of the article is that the authors were unable to identify the causative agent of the disease, if any. It is possible that this disease is not pathogenic in nature, but is determined by a violation of metabolic processes associated with deficiencies in cultivation (inadequate food, stress, increased housing density etc.). It worth discussion.
Some sentences appear to be grammatically incorrect: see for example lines 25 and 35.
Author Response
Reviewer 1:
The main comments are as follows:
- Text should be checked for misspellings, see for example genus name in line 35. When a species is first mentioned, it is necessary to also indicate the authors and year: Oncorhynchus mykiss 11.
R: The text has been edited, an including in the title.
- In the abstract and text there is clearly an excessive number of abbreviations for the name of the disease mentioned. Only in the abstract the abbreviation SD is given three times. It is enough to do this once in the abstract and once at the first mention in the text.
R: This was corrected text has been edited
- The abstract is unjustifiably verbose; it contains information, for example, the place of which is in materials and methods (an indication of weight of fish). The text of the abstract should be significantly reduced, leaving only the most essential.
R: This section has been edited. Changes were marked in yellow.
The main drawback of the article is that the authors were unable to identify the causative agent of the disease, if any. It is possible that this disease is not pathogenic in nature, but is determined by a violation of metabolic processes associated with deficiencies in cultivation (inadequate food, stress, increased housing density etc.). It worth discussion.
R: We agree with the reviewer regarding the fact that we were not able to identify the actual etiology of the disease; however, we need to stress that the main goal of the present study was not that one, and thus our experimental design did not look to answer that question; currently, we are unable to fulfill Koch`s postulates, particularly to exogenously culture the pathogens, in order to identify the actual pathogen. Our main goal, as mentioned along the manuscript, was to identify and described potential differences in the composition of the skin-associated microbiota between fish with and without external Strawberries disease-derived skin lesions. By doing that, we believe we contribute to better understand potential dysbiosis occurred in skin-associated microbiota, which can help us to further analyzed lesion progression and changes in skin microbiota in future research.We have discussed the possibility that this disease is only confined to cultured trout, as suggested by the reviewer in the last paragraph of the discussion section.
Some sentences appear to be grammatically incorrect: see for example lines 25 and 35
R: we have edited the sentences in line 25 and 35, as suggested by the reviewer.
Reviewer 2 Report
Comments and Suggestions for Authors
The aim of this manuscript is to compare the composition of the skin-associated microbiota of healthy rainbow trout with that of skin of trout suffering SD lesions, using an RNA-based approach which enables the identification of the active bacterial population. Since this is the real novelty of the study, instead of saying that this is the first description of microbiota associated to SD injured-skin samples, it would be more appropriate to add "using an RNA-based approach".
There are many works in the literature on the histopathology of skin samples with SD lesions. Therefore, to provide something new and improve the manuscript, I suggest to analyze the changes of the microbiota diversity at different stages of lesion progression.
In lines 340-343, the authors should only suggest that Candidatus Midichloria "seems to play an important role as etiological agent of SD."
Although I am not a native English speaker, some errors detected are:
line 28, change revealing by revealed
line 36, delete the second "is"
line 375, change This by These
Author Response
Reviewer 2:
The aim of this manuscript is to compare the composition of the skin-associated microbiota of healthy rainbow trout with that of skin of trout suffering SD lesions, using an RNA-based approach which enables the identification of the active bacterial population. Since this is the real novelty of the study, instead of saying that this is the first description of microbiota associated to SD injured-skin samples, it would be more appropriate to add "using an RNA-based approach".
R: Thanks, we included the suggestion.
There are many works in the literature on the histopathology of skin samples with SD lesions. Therefore, to provide something new and improve the manuscript, I suggest to analyze the changes of the microbiota diversity at different stages of lesion progression.
R: Thanks. In this work, we compare the extremes: healthy versus severe lesions. We are working in the different stages of lesions, we hope this will be our new manuscript.
In lines 340-343, the authors should only suggest that Candidatus Midichloria "seems to play an important role as etiological agent of SD."
R: Thanks, this was corrected.
Although I am not a native English speaker, some errors detected are:
line 28, change revealing by revealed
line 36, delete the second "is"
line 375, change This by These
R: Those errors were corrected